# A Novel Green Extraction Technique for Extracting Flavonoids from *Folium nelumbinis* by Changing Osmosis Pressure

**DOI:** 10.3390/ma13184192

**Published:** 2020-09-21

**Authors:** Hai-Yan Fang, Ying-Qin Wei, Meng-Li Zhang, Wei Liu

**Affiliations:** 1School of Bioengineering, Qilu University of Technology (Shandong Academy of Sciences), No. 3501, Daxue Road, Changqing District, Jinan 250353, China; fhy@qlu.edu.cn; 2Key Laboratory of Fine Chemicals in Universities of Shandong, School of Chemistry and Pharmaceutical Engineering, Qilu University of Technology (Shandong Academy of Sciences), No. 3501, Daxue Road, Changqing District, Jinan 250353, China; zhangmengli@qlu.edu.cn (M.-L.Z.); liuw2029592485@qlu.edu.cn (W.L.)

**Keywords:** osmosis extraction, inorganic salt, flavonoid, *Folium nelumbinis*, aqueous two-phase system

## Abstract

A new green and sustainable extraction technique, namely osmosis extraction (OE), was developed for efficient extracting flavonoids from *Folium nelumbinis* by changing the osmotic pressure. The antioxidant activities of the extracted flavonoids were also evaluated. Ethanol and ammonium sulfate were selected for the OE system because they are environmentally friendly. The maximum flavonoids concentration in the top phase was obtained with an ethanol volume fraction of 42.0% and the salt mass of 1.9 g. The kinetic behavior of the extraction process showed that OE had higher efficiencies especially coupled with ultrasonication due to the accompanying and serious morphological changes of *Folium nelumbinis* cells observed by digital microscope and nano-computed tomography (nano-CT). Results of morphological and anatomical features showed that the higher intracellular chemical potential made the cell expand and even led to bursting. The results also showed that the extraction efficiency of flavonoids with high antioxidant activities was higher than that of the traditional method. The interface effect enhanced the extraction during the salting-out extraction and osmosis was the main factor that improved the extraction efficiency.

## 1. Introduction

A new green and sustainable extraction technique, namely osmosis extraction (OE), was first developed based on salting-out extraction (SOE). SOE has attracted much attention recently because it is easy to scale up, inexpensive, and is a green technique [1,2]. The extraction of target products from complex material was based on their selective distribution between the aqueous two-phase system. The extraction system usually contains low molecular weight organic solvents, such as methanol, ethanol, acetonitrile, acetone, and *n*-propanol, and inorganic salts. SOE techniques have been applied to extract active components such as flavonoids [3,4], alkaloids [5], phenolic compounds [6], polysaccharides [7], lignans [8], phenylethanoid glycosides [9], ginsenosides [10,11], and succinic acid [12] from biological resources. In addition, SOE techniques have an attractive application prospect for separation and purification of active proteins, such as lipase [13], phage [14], serine protease [15], etc. As far as we know, the SOE technique has usually been used to deal with liquid materials and few reports have involved the extraction of solid materials. Lotus, as a common vegetable, is also an important medicinal plant. Lotus leaves and their extracts have been developed into various food supplements due to their important biological activities such as anti-oxidant [16], anti-osteoporosis [17], anti-viral [18], anti-obesity [19], hepatoprotective [20], etc. In addition, *Folium nelumbinis*, the dry leaves of *Nelumbo nucifera* Gaertn which are commonly harvested in summer and autumn, have a wide range of clinical applications in traditional Chinese medicine. Chemical analysis showed that its chemical composition was mainly composed of flavonoids, polysaccharides, alkaloids, and so on. Among them flavonoids have attracted much attention recently due to their significant antioxidant activity. Some extraction techniques were used to extract flavonoids from dry leaves such as the water decocting method [21], enzyme-assisted extraction [22], etc. However, the extraction solvent is difficult to infiltrate into the medicinal materials and the medicinal components are difficult to dissolve and diffuse out of the material cells. During the SOE process for extracting herbal materials, it was found that promotion of dissolution by inorganic salts could increase the extraction efficiency [3]. In fact, the top and bottom phases have different compositions, especially when there is a large difference in the osmotic pressure between the top and bottom phases. In this paper, the effect of inorganic salts on promoting dissolution was explored. In addition, the mechanism of promoting dissolution was elucidated by the morphological changes of *Folium nelumbinis* cell observed by digital microscope and nano-computed tomography (nano-CT) for the first time. Then the new green and sustainable extraction technique was developed based on the above experimental data.

## 2. Experimental

### 2.1. Materials and Reagents

*Folium nelumbinis* samples were purchased from a local drugstore (Jinan, China) and ground to a fine powder in a micro plant grinding machine (Shanghai, China). The powdered material was sieved through 40 mesh nylon gauze. The fraction larger than 40 mesh was collected and powdered again until the powder passed through the 40 meshes sieve. Phosphoric acid, ethanol, ammonium sulfate, and other chemicals were of analytical grade and purchased from Chinese Pharmaceutical Group Tianjin Chemical Reagent Company (Tianjin, China). Flavonoid standards of hyperoside, rutin, and quercetin were procured from the National Institute for Food and Drug Control (Beijing, China). Acetonitrile and methanol were of high-performance liquid chromatography (HPLC) grade (Tedia Co., Inc., Fairfield, OH, USA). The water-purification system from Millipore Co. (Billerica, MA, USA) was used.

### 2.2. Apparatus

The analysis was carried out on a Shimadzu HPLC (Kyoto, Japan) equipped with a dual LC-20A pump, SPD-20A ultraviolet–visible (UV–vis) detector, and an auto-injector. The chromatograms were integrated using a LC-solution chromatographic station. A UV spectrophotometer (Spectronic UV-160A, Shimadzu, Roucaire, Courtaboeuf, France), AE 240 electronic balance (0.01 mg/20 g, Mettler Toledo, Zurich, Switzerland), Nikon Microscope Eclipse E100 (Nikon Co., Tokyo, Japan), Nano CT (SkyScan 2211, Bruker, Antwerpen, Belgium), JFSD-100 micro plant grinding machine (Shanghai, China), and KQ 220 ultrasonic cleaner (Kunshan Ultraso Instrument Co., Ltd., Kunshan, China) were used.

### 2.3. Chromatographic Conditions

Anasil AQ-C_18_ column (150 mm × 4.6 mm i.d., 5 µm, Shimadzu, Kyoto, Japan) was used to analyze flavonoids at a column temperature of 30 °C. A gradient elution with a binary solvent system of methanol (A) and 0.1% phosphoric acid solution (B) was used. The gradient elution was conducted as follows: 0.01–8 min, 10–40% A; 8.01–23 min, 40–70% A; and 23.01–30 min, 70–85% A. The mobile phase flow rate and detector wavelength were set at 1.0 mL/ min and 360 nm respectively.

### 2.4. Standard Solution Preparation

Standard solution of hyperoside, rutin, and quercetin were prepared in methanol respectively. Then a series of solutions with different concentrations were obtained by diluting the above standard solutions.

### 2.5. Flavonoid Extraction Method

#### 2.5.1. Osmosis Extraction

About 0.2 g of *Folium nelumbinis* powder was weighed precisely and extracted by osmosis extraction (OE) with ethanol and ammonium sulfate solution for a set time at room temperature. After extraction and phase separation, the upper and lower phases were collected, filtered, and analyzed to determine the flavonoids concentrations.

#### 2.5.2. Ultrasonication-Assisted Osmosis Extraction (OE)

The extraction conditions were the same as those for the OE (Section 2.5.1). The only difference was that the sample was ultrasonicated at 100 W and 40 KHz during the extraction.

#### 2.5.3. Single-Phase Extraction

A flask containing 0.2 g of Folium nelumbinis powder was extracted with 20 mL of extraction solvent (60% ethanol/water) for 0.5 h. The extract was filtered, transferred to a 25-mL volumetric flask, and the volume was made up to the mark with the extraction solvent to prepare a test solution.

#### 2.5.4. Ultrasonication-Assisted Single-Phase Extraction

The extraction conditions were the same as for the single-phase extraction (Section 2.5.3). The only difference was that the sample was ultrasonicated at 100 W and 40 KHz during the extraction.

### 2.6. Total Flavonoid Content Determination

The aluminium nitrate colorimetric method [23] with rutin as a standard was used to determine the content of total flavonoids. The test solution (1.5 mL) and a 5% sodium nitrite solution (0.25 mL) were mixed well and then set aside for 6 min. Then, 0.25 mL of a 10% aluminum nitrate solution was added, and the solution was mixed and set aside for another 6 min. Next, 2.5 mL of sodium hydroxide solution was added with shaking, followed by addition of 1.75 mL of water with shaking, and the solution was left to stand for 15 min at room temperature. Finally, the solution was analyzed at 510 nm and compared with a blank containing all reagents and solvents but not the test sample. The test solution was filtered before HPLC analysis.

### 2.7. Antioxidant Activity Against 2,2-Diphenyl-1-Picrylhydrazyl Radicals

Antioxidant activity was assayed using 2,2-diphenyl-1-picrylhydrazyl (DPPH, Purity ≥ 98.0%, Cool Chemistry, Beijing, China). Aliquots (1.0 mL) of the sample solutions or blank solvents and 2.0 mL of 0.0045% DPPH/ethanol solution were mixed and then left at room temperature. After 30 min, the absorbance of the test solution was measured at 517 nm using ethanol as a blank control and quercetin as a positive control. The radical scavenging activity was measured as the decrease in the absorbance (A) of DPPH and calculated using the following equation: inhibition (%) = [1 − (A_i_ − A_j_)/A_0_] × 100%. The subscripts of A such as i, j, and 0, referred to tested solution, reagent blank and reference solution, respectively.

### 2.8. Statistical Analysis

All experiments were performed twice and the reported values are averages. The experimental data were analyzed using Statistical Product and Service Solutions (SPSS, IBM, New York, USA) and Origin 6.0 software (OriginLab, Northampton, MA, USA).

## 3. Results and Discussion 

### 3.1. Validation of the Analytical Methods

For the colorimetric method to determine total flavonoids, rutin was used as a standard and the calibration curve range was 0.064 to 0.24 mg/mL with good linearity (*r* = 0.9993). The main flavonoids were separated and analyzed using HPLC with a gradient elution (Figure 1).

The results showed that the total flavonoids were made up of 42.65% hyperoside and 0.79% quercetin, which were obtained with high recoveries of 97.52% and 92.84%, respectively.

### 3.2. Optimization of the OE Conditions

SOE systems, such as ammonium sulphate/ethanol, sodium dihydrogen phosphate/ethanol, have been commonly used to extract bioactive compounds [24]. According to the relative Food and Drug Administration (FDA) act, these systems are generally recognized as safe (GRAS) since they were environmentally friendly. It was found that an ethanol/(NH_4_)_2_SO_4_ system provided good extraction of flavonoids. Ethanol was selected as the organic solvent for the extraction system because it is inexpensive, nontoxic, and dissolves flavonoids. The ethanol/(NH_4_)_2_SO_4_ system is an ideal green system for **OE**. The extraction efficiencies of the flavonoids were investigated by changing the proportions of ethanol and inorganic salts in the ethanol/(NH_4_)_2_SO_4_ system. Flavonoids were found to distribute more in the top phase of extraction system than the bottom phase.

The phase diagram of OE with the ethanol/(NH_4_)_2_SO_4_ system at 15 ± 4 °C (Figure 2) was plotted using the turbidity point method. 

The upper part of the curve was the region of two aqueous phases and the lower part was a single-phase region. The composition of upper and lower phase of the two aqueous phase system was significantly different, so that the solute could be selectively distributed in one phase.

The addition of ethanol to a concentrated saline aqueous solution led to formation of an ethanol-rich upper phase and a salt-rich lower phase.

#### 3.2.1. Effect of Ethanol Concentration

Ethanol was added to a solution containing 1.8 g of (NH_4_)_2_SO_4_ for OE; meanwhile, the volume of the extraction system was maintained about 9 mL. Then, 0.2 g of *Folium nelumbinis* powder was added and the sample was extracted for 1.5 h. After extraction and phase separation, the upper phase was collected, analyzed (Section 2.6), then the volume ratio of upper and bottom phase, absorption of test solution was obtained. Results showed that extraction efficiency was obviously influenced by the ethanol volume fraction in the extraction system (Table 1).

As the ethanol volume fraction decreased, the flavonoids extraction efficiency first increased, then went down, and then increased again slightly. The maximum concentration of flavonoids (0.05 mg/L) extracted was obtained with an ethanol volume fraction of 42%. The volume ratio of the top to bottom phase showed a similar trend to the flavonoid concentration. Thus, an ethanol volume fraction of 42% was selected for use in subsequent experiments.

#### 3.2.2. Effect of the Mass of Salt

The effect of the mass of salt on the extraction efficiency was explored while holding the other conditions in the OE system constant. The flavonoids extraction efficiency increased as the mass of salt increased from 1.6 to 1.9 g and reached a maximum value with 1.9 g of salt (Table 2).

When the mass of salt exceeded 1.9 g, an aqueous two-phase system did not form easily under the current experimental conditions. The flavonoids concentration and phase volume ratio mainly varied with increases in the salt mass. Changes in the system pH had little effect, although the system pH could be influenced by salt mass, for example, the pH values of 15% and 18% ammonium sulfate solutions are 4.34 and 4.02, respectively. From these results, 1.9 g of (NH_4_)_2_SO_4_ was selected for use in the OE system.

#### 3.2.3. Kinetics of the Extraction Process

The extraction time could greatly affect the extraction equilibrium and kinetics of the extraction process. The extraction efficiency with the proposed method (Figure 3a,b) and conventional extraction method (Figure 3c,d) were compared by analyzing samples taken at different time points in the extraction process.

The proposed method had a higher extraction efficiency than the conventional method. The extraction time greatly affected the extraction equilibrium because of the time-consuming dissolution step. The difference in extraction efficiency was greater with longer extraction times and ultrasonication had a large effect on the extraction equilibrium (Figure 3a,c). For the ultrasonication-assisted single solvent extraction method, the flavonoids’ extraction efficiency increased with the extraction time and reached a maximum value within 50 min. By contrast, without ultrasonication, the flavonoids maximum value was not reached until 100–120 min. These observations could be explained by the difference in intracellular osmotic pressure when using saline and non-saline solvents, and suggests that the inorganic salt promotes dissolution during the extraction process. Additionally, ultrasonication could help with breaking down cell walls and hastening dissolution of the flavonoids. In fact, the whole OE process could be divided into two steps: the dissolution step, and the two-phase extraction step. The dissolution process was the first step and the rate-controlling step. In the two-phase extraction step, the transfer of flavonoids was very fast and the extraction equilibrium could be achieved quickly. Therefore, the most time-consuming step for extracting flavonoids was the dissolution process. The extraction efficiency increased slightly with increases in the extraction time after a certain period (Figure 3d). An extraction time of 50 min was used to extract flavonoids from the herbal material when using ultrasonication. 

#### 3.2.4. Mechanism of the Promotion of Dissolution by Inorganic Salts

The mechanism of the promotion of dissolution by the inorganic salts was explored from the points of view of the extraction process and the cell structure. To observe the anatomical properties of the structural deformation, a digital microscope and nano-CT were used to obtain morphological and anatomical features of the drug material by different pretreatment methods. In the dissolution step, swelling and the fracture of plant cells determined the extraction speed. Compared with single-phase extraction (Figure 4a), OE (Figure 4c) resulted in morphological changes to the surface structure of leaves and obvious holes were observed.

The leaf cells of *Folium nelumbinis* after OE were generally larger than those after the single-phase extraction, and there seemed to be many holes in the cell walls (Figure 4a–d). This indicated that the cell walls had burst and the cells had fused. Further magnification (Figure 4a’–d’) showed no complete cells, which indicated that the cell walls were broken and the cell contents had leaked out. The OE treated leaves displayed visible structural damage with the formation of pores and gaps. Nano-CT anatomy of the leaf plane fault also confirmed this, and compared with traditional extraction methods, cell damage by OE was so severe that no complete cell could be seen (Figure 5a,b).

Sectional anatomy was more obvious since only the upper and lower epidermis and a small number of veins of the leaf were visible while mesophyll tissue was missing (Figure 5c,d). A nano-CT image of a Lotus leaf demonstrated that the dramatic changes in leaf structure after OE could facilitate penetration of the solvent into the cell interior easily, and promoted dissolution and diffusion of some active components to the outside of the cell lightly, which improved the extraction effect obviously. The water channel protein aquaporin could adjust the intracellular osmotic pressure. The extraction solvent with a high salt concentration gave a higher intracellular than extracellular chemical potential. The extraction solvent quickly crossed the cell wall and membrane and entered the cell, which made the cell expand and even led to bursting (Figure 4). This difference between the cells could be attributed to the difference in osmotic pressure in the medium. Serious damage to the cell walls was observed after the ultrasonication-assisted extraction (Figure 4b,d). The ultrasonication-assisted extraction could greatly improve the extraction efficiencies of active components from medicinal materials. Moreover, the formation of an interface in the two-phase extraction system could result in an interface effect with the two-phase extraction system. The interface effect could promote the breaking of cell walls and release of medicinal compounds into the top and bottom phases, which would greatly improve the extraction efficiency. The difference in osmotic pressure was the main factor that improved the extraction efficiency, and ultrasonication could enhance the extraction effect through cell disruption.

### 3.3. Evaluation of the Proposed Separation Process

The yield, purification factor (PF) and activity were evaluated for the proposed separation process and conventional separation process. The yield (%), PF of flavonoids (%) and PF of hyperoside (%) were calculated using the following equations: yield (%) = extract amount/drug material amount × 100%, PF of flavonoids (%) = flavonoids amount/extract amount × 100%, and PF of hyperoside (%) = hyperoside amount/extract amount × 100%. The ability to scavenge DPPH radicals was evaluated using the 50%inhibiting concentration (IC_50_, μg/mL). The proposed method, SOE method, and resin method were compared for flavonoid extraction (Table 3).

Among these methods, the proposed method had a higher extraction recovery rate, higher purification factors of flavonoids and hyperoside, and higher scavenging activity. The SOE and resin methods gave similar results. The enhanced extraction performance could be attributed to cell rupture and leakage of cell content.

## 4. Conclusions

An OE technique was developed using the SOE procedure to enhance the extraction efficiency of pharmaceutical components from herbal materials by changing the osmotic pressure. The intracellular osmotic pressure differed with saline and non-saline solvents. The phase interface, which existed in the OE biphasic system, could promote rupture of the cell walls of medicinal materials. This greatly improved the extraction efficiency. In addition, the effect of osmotic pressure was verified by studying structural changes in the cell wall and cell membrane of *Folium nelumbinis*. In summary, OE is promising as a green extraction method that is rapid, efficient, and has low energy consumption.

## Figures and Tables

**Figure 1 materials-13-04192-f001:**
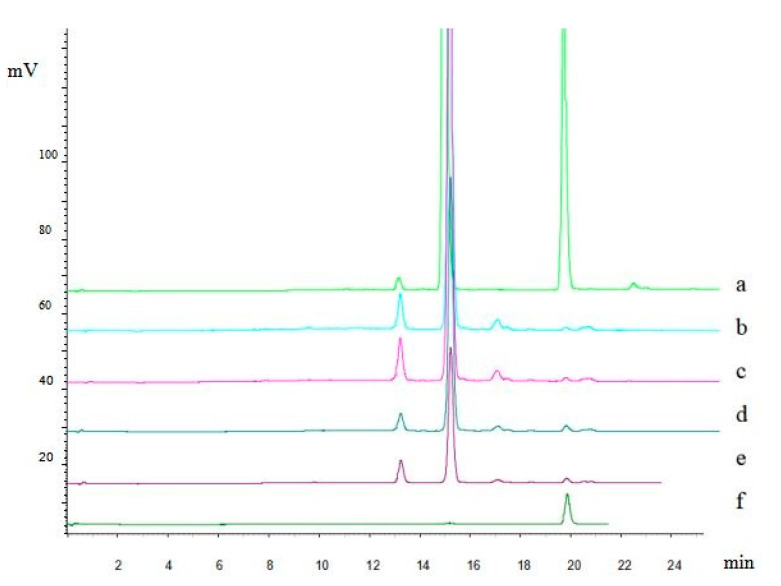
Chromatograms of different batch of flavonoids from Folium nelumbinis obtained by the osmosis extraction method (equivalent raw material 6 mg ml^−1^). (**a**)—Mixed standard solution of hyperoside and quercetin; (**b**)—flavonoids (batch No. 160525); (**c**)—flavonoids (batch No.160528); (**d**)—flavonoids (batch No.160523); (**e**)—flavonoids (batch No.160528); (**f**)—standard solution of quercetin.

**Figure 2 materials-13-04192-f002:**
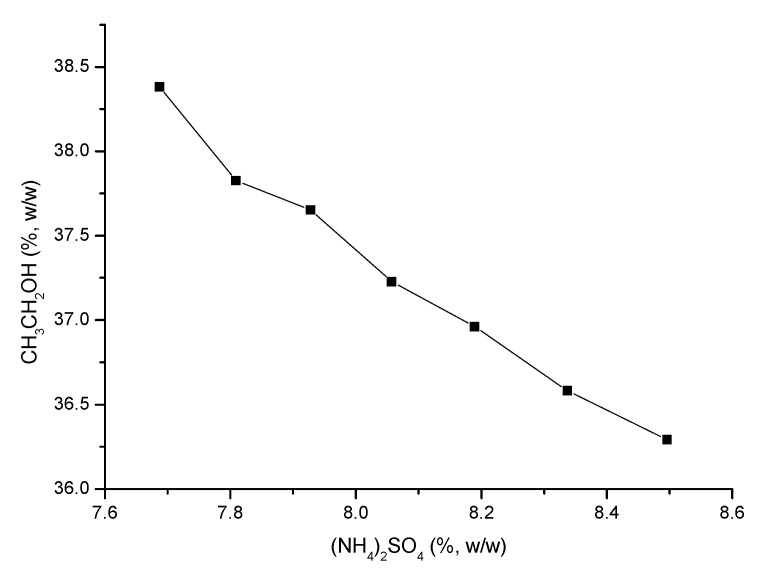
The phase diagram of osmosis extraction (OE) with the ethanol/(NH_4_)_2_SO_4_ system.

**Figure 3 materials-13-04192-f003:**
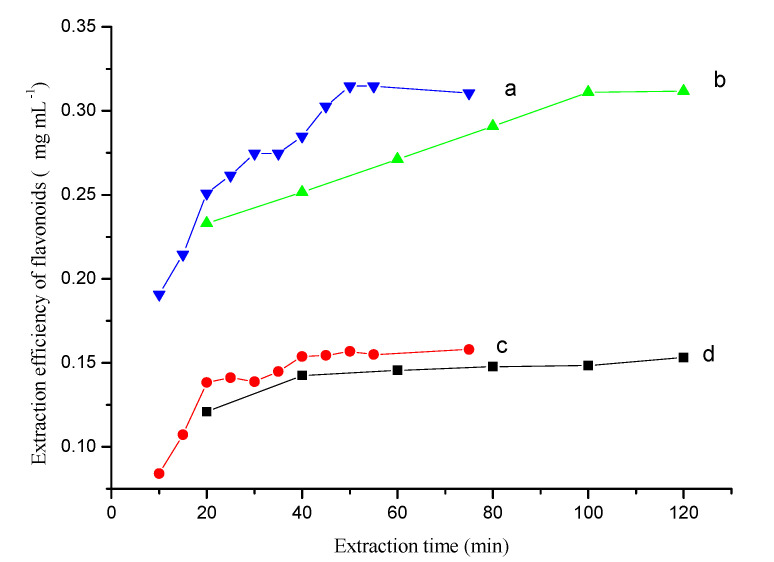
The kinetic behavior of extraction process obtained by the proposed (**a**,**b**) and traditional extraction method (**c**,**d**). (**a**)—Ultrasonic-assisted osmosis extraction, (**b**)—Osmosis extraction, (**c**)—Ultrasonic-assisted single solvent extraction, (**d**)—Single solvent extraction. Conditions, Folium nelumbinis powder, 0.2 g.

**Figure 4 materials-13-04192-f004:**
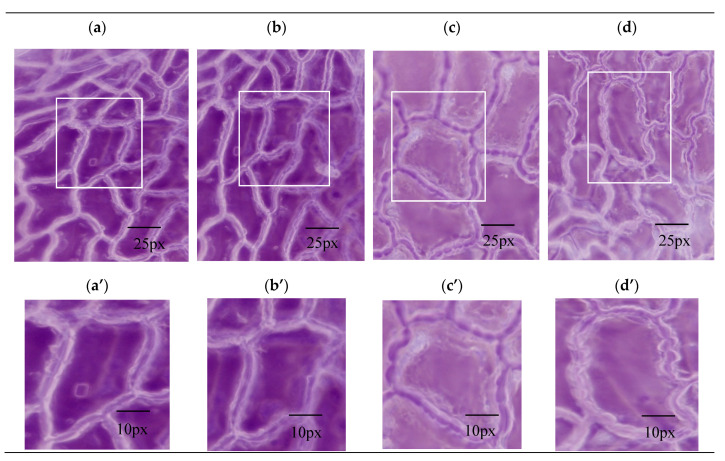
The photomicrograph (**a**–**d**) and magnified montages of boxed regions in images above (**a’**–**d’**) of *Folium nelumbinis* by different extracting methods. Scope Image 9.0, magnification times: 400×. (**a**,**a’**)—Single-phase extraction; (**b**,**b’**)—Ultrasound-assisted single-phase extraction; (**c**,**c’**)—Osmosis extraction; (**d**,**d’**)—Ultrasound-assisted osmosis extraction.

**Figure 5 materials-13-04192-f005:**
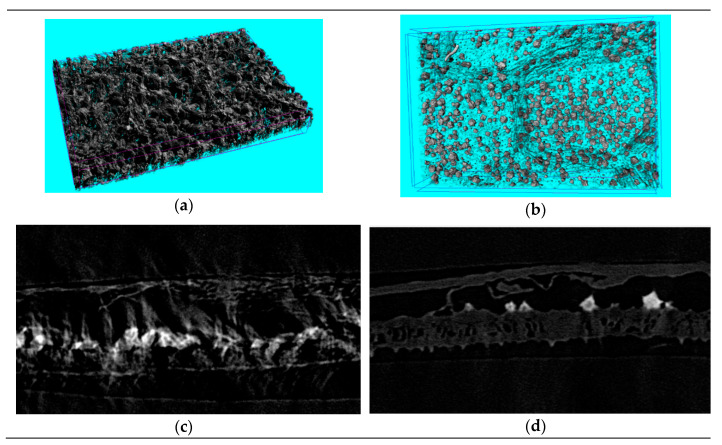
The nano-computed tomography (nano-CT) anatomy of plane fault (**a**,**b**) and sectional anatomy of Folium nelumbinis (**c**,**d**) treated by different extracting methods. (**a**,**c**)—Single-phase extraction, (**b**,**d**)—Osmosis extraction.

**Table 1 materials-13-04192-t001:** Effect of the ethanol concentration on flavonoid extraction.

Sample No.	Ethanol (mL)	Volume Ratio	Absorption	Flavonoids (mg/L)
1	4.0	1.714	0.492	0.043
2	3.8	2.000	0.577	0.050
3	3.5	1.375	0.375	0.0325
4	3.3	1.594	0.345	0.030
5	3.0	0.750	0.469	0.041

**Table 2 materials-13-04192-t002:** Effect of the mass of salt on flavonoid extraction.

Sample No.	Mass of Salt (g)	Volume Ratio	Absorption	Flavonoids (mg/L)
1	1.6	1.771	0.253	0.044
2	1.7	2.214	0.276	0.048
3	1.8	2.000	0.307	0.054
4	1.9	1.400	0.321	0.056
5	2.0	_		

**Table 3 materials-13-04192-t003:** Evaluation of the proposed separation process.

Method	Yield (%)	PF of Flavonoids (%)	PF of Hyperoside (%)	IC_50_ (μg/mL)
This method	4.20	85.71	61.11	6.4
SOE method	3.75	86.13	46.44	6.2
Resin method	3.49	86.53	47.35	6.2

PF—purification factor (PF); IC_50_—50%inhibiting concentration.

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
