# Peer review of "A Novel Green Extraction Technique for Extracting Flavonoids from Folium nelumbinis by Changing Osmosis Pressure"

_materials, 2020, doi:10.3390/ma13184192_

Round 1
Reviewer 1 Report
Authors are claiming to present a new extraction technique. In order to convince this reviever, the next questions and comments should be answered:
- The introduction does not give enough information of the background of the technique and the plant, extracts. Also, a schematic figure for the instrumental setup would help the reader in understanding.
- References are requested supporting the importance of this plant.
- The novelty of the technique should be supported by more facts.
- What was the wavelength used for the integration of the HPLC chromatogram?
- Ai, Aj, A0, Vr, A should be explained in the text or in captions
- The % of flavonoids reported corresponds to each or the sum or the average of the samples?
- How did the authors find this mixture (i.e. EtOH and ammonium sulfate)? What else could be used or couldn't?
- The changes in the pH should be defined and numbers should be shown (page 6)
- The Nano-CT pictures are hard to evaluate and analyze. There are proposed explanations in the text, but this reviewer could not really see what the authors claim on the figures. I suggest to revise this section, and probably add some explaining highlighting, text boxes etc. to the figures.
- uthors claim that the cells became larger and there are many holes. It should be quantified and pointed at the figure.
Author Response
Dear Sir,
Manuscript ID materials-918018 entitled " A novel green extraction technique for extracting flavonoids from Folium nelumbinis by changing osmosis pressure" has been revised carefully according to the reviewer's suggestions.

Reviewer 2 Report
Referee report for the manuscript molecules-918018.v1:
„A novel green extraction technique for extracting flavonoids from Folium nelumbinis by changing osmosis pressure”
by Hai-Yan Fang, Ying-Qin Wei, Meng-li Zhang and Wei Liu
The authors describe a novel extraction technique for the isolation of phenolic compounds from plants, based on using an environmentally friendly organic solvent (ethanol) and cell membrane permeabilization/disruption by salt addition.
The proposed approach is interesting as it combines two steps simultaneously that are normally only performed sequentially in an extraction procedure. Yet, this makes it difficult to understand which of the two parts is the more relevant for an efficient extraction. The manuscript is sufficiently well written in most cases, although it occasionally needs some lingual improvement, notably in the introduction.
The experimental part is clear; however, the results and discussions section fails in part in convincingly relating the experimental results to their interpretation: As a consequence, not all of the findings of the authors appear to be supported by the experimental data to that extent that the authors claim.
Detailed comments and suggestions:
- Introduction section: needs occasional lingual improvement (e.g., …which harvested… / …was difficult to infiltrate… / …inorganic salts could increasing…)
- Apparatus: be consistent ! If you name the supplier, then it should be: company name, place, country.
- Figure 1: The figure is unsuitable for reproduction: lines are too weak; lettering is too small; headers in grey are unreadable and shall be removed. Time scale and axis label overlap.
- Figure 2: It is not clear why/how this figure is a phase diagram. Improve visual presentation and explain what do you mean by this diagram !
- Table 1: “Vr” and “A” are not explained in the text nor in the table legend. If the percentage of ethanol is such an important aspect, then it should also be reported directly in the table. Furthermore, I disagree with the statement that the flavonoids extraction efficiency first increases and then decreases. If one examines the values in Table 1, then they first increase (0.043 -> 0.05), then go down (0.05 -> 0.03), but then increase again (0.03 -> 0.041). This should be correctly reported in the text.
- Table 2: Also in this table, “Vr” and “A” are not explained.
- In the discussion of section 3.2.3, it is not entirely clear whether the “dissolution” process (considered as the rate-limiting step) is not rather the cell disruption step (as a prerequisite for efficient extraction).
- The micrographs in Figure 4 are not so clear to me as to allow only the authors’ interpretation of cell volume increase and cell wall disruption. Also, the nano-CT is not sufficiently well described/interpreted to unambiguously support the claims of the authors why OE is more efficient than regular extraction, particularly when performed under ultrasonication.
- Table 3: What is the “Purification factor” ? Explain ! Also please note that it is not adequate to report percentage values with two digits after the decimal point while no statement is made of the reproducibility of measurements (which is much higher, I assume).
- Conclusion: I do not see the point why it is the “phase interface, which could promote rupture of the cell walls of medicinal materials”. I believe that the key is that under the combined action of EtOH solvent and salt the cell walls are disintegrated, and hence extraction may proceed more successfully.
- Funding: “polisiting” is probably “polishing” ?
- References: The format of the references is inconsistent. Initials are abbreviated with or without dots; page nos. are given as start page only or as start-end page; volume numbers are reported differently. Ref. 11+12 apparently is only one reference.
Apart from all indicual points mentioned above, a more general question to raise is whether "Materials" is the best suitable journal to publish this manuscript.
Author Response
Dear Sir,
Manuscript ID materials-918018 entitled " A novel green extraction technique for extracting flavonoids from Folium nelumbinis by changing osmosis pressure" has been revised carefully according to the reviewer's suggestions. The main changes are as follows.

Reviewer 3 Report
The authors describe a "green" method to extract flavonoids from Folium nelumbinis using a newly developed technique called osmosis extraction (OE). The authors compared this using more the more traditional technique Salting Out Extraction (SOE). Furthermore, the authors explored means to increase the rate of extraction as well as provided a viable mechanism for the improvements seen in the new extraction method. It is this reviewer's opinion that this should be published with some minor revisions listed below:
- Overall, some minor editing for English should be performed. It is recognized that English is not the authors' first language, but some minor editing would make portions of this manuscript clearer.
- Page 2, Section 2.1: The authors' state "Folium nelumbinis samples were purchased from a local drugstore (Jinan, China) and powdered (40 mesh)." How were the samples powdered and was the size (40 mesh) simply assumed to be the size as this was the last filter size employed? A short description of the process would be helpful to readers not versed in this type of work.
- Page 3, Section 2.6: The authors should add a reference for the aluminium nitrate colorimetric method.
- Page 3, Section 2.7: The authors should add a reference for the inhibition equation.
- Page 4, Figure 1: This figure is difficult to read, especially the axis names and the information at the top. It is not very clear if that information is needed. It may make this clearer if the data is extracted and plotted in another software program.
- Page 4, Section 3.2: The authors should add a reference for the turbidity point method.
- Pages 5-6, Tables 1 and 2: The abbreviations VR and A must be defined or referenced in the text. Without these specifically defined, it is difficult to follow the narrative describing the results in the table. It can be inferred that VR is the Volume Ratio, but what is A? Is it the percentage of one of the layers in VR?
- Pages 7-8, Figure 4: This figure should be significantly modified to enhance clarity. First, the first column (Image Properties) needs to be modified to ensure that words do not split over multiple lines. Second, for the second row, should the labels be a', b', c', d'? Finally, it may make more sense to move the last two rows (the Nano-CT results) to another figure.
- Tables 1, 2, and 3: The column titles need to be modified to ensure that words and units do not split over multiple lines.
Author Response

(The authors gave the same response as above.)

Round 2
Reviewer 1 Report
This reviewer accepts the answers of the authors, but still misses the exact length-scale from Fig. 4 (e.g. ____ 100 nm).
Author Response
Dear Sir,
The exact length-scale had been added to the Figure 4 according to the instruction.
